# Regulation of Aqueous Humor Secretion by Melatonin in Porcine Ciliary Epithelium

**DOI:** 10.3390/ijms24065789

**Published:** 2023-03-17

**Authors:** Ka-Lok Li, Sze-Wan Shan, Fang-Yu Lin, Choi-Ying Ling, Nga-Wai Wong, Hoi-Lam Li, Wei Han, Chi-Ho To, Chi-Wai Do

**Affiliations:** 1School of Optometry, The Hong Kong Polytechnic University, Hong Kong, China; 2Centre for Eye and Vision Research (CEVR), 17W Hong Kong Science Park, Hong Kong, China; 3Research Centre for SHARP Vision (RCSV), The Hong Kong Polytechnic University, Hong Kong, China; 4Department of Ophthalmology, Zhejiang University, Hangzhou 310027, China; 5Department of Ophthalmology, Emory University, Atlanta, GA 30322, USA; 6Research Institute of Smart Ageing (RISA), The Hong Kong Polytechnic University, Hong Kong, China

**Keywords:** melatonin, ciliary epithelium, gap junction, aqueous humor formation, glaucoma

## Abstract

Secretion of melatonin, a natural hormone whose receptors are present in the ciliary epithelium, displays diurnal variation in the aqueous humor (AH), potentially contributing to the regulation of intraocular pressure. This study aimed to determine the effects of melatonin on AH secretion in porcine ciliary epithelium. The addition of 100 µM melatonin to both sides of the epithelium significantly increased the short-circuit current (Isc) by ~40%. Stromal administration alone had no effect on the Isc, but aqueous application triggered a 40% increase in Isc, similar to that of bilateral application without additive effect. Pre-treatment with niflumic acid abolished melatonin-induced Isc stimulation. More importantly, melatonin stimulated the fluid secretion across the intact ciliary epithelium by ~80% and elicited a sustained increase (~50–60%) in gap junctional permeability between pigmented ciliary epithelial (PE) cells and non-pigmented ciliary epithelial (NPE) cells. The expression of MT_3_ receptor was found to be >10-fold higher than that of MT_1_ and MT_2_ in porcine ciliary epithelium. Aqueous pre-treatment with MT_1_/MT_2_ antagonist luzindole failed to inhibit the melatonin-induced Isc response, while MT_3_ antagonist prazosin pre-treatment abolished the Isc stimulation. We conclude that melatonin facilitates Cl^−^ and fluid movement from PE to NPE cells, thereby stimulating AH secretion via NPE-cell MT_3_ receptors.

## 1. Introduction

Glaucoma is a sight-threatening disease characterized by a gradual and permanent loss of retinal ganglion cells, resulting in progressive loss of vision. Although the pathogenesis of glaucoma is not fully understood, elevated intraocular pressure (IOP) remains as the only modifiable risk factor identified thus far. IOP is dictated by the rates of aqueous humor (AH) secretion and drainage. The ciliary epithelium, which is composed of pigmented ciliary epithelium (PE) and non-pigmented ciliary epithelium (NPE) layers, actively secretes AH. After that, AH exits the eye primarily through the conventional pressure-dependent trabecular meshwork (TM) outflow pathway. Active Cl^−^ transport across the ciliary epithelium has been shown to be the major driving force for AH secretion [1,2]. The intercellular gap junctions at the apical surfaces of PE and NPE cells link these two cell layers to work in a coordinated manner as a syncytium [3,4,5]. In addition, swelling-activated Cl^−^ channels located at the basolateral membrane of NPE cells are thought to likely play important roles in regulating AH secretion rates via Cl^−^ release [3,6,7].

Melatonin, a natural hormone potentially regulating AH dynamics, is an indole-derived compound primarily secreted by the pineal gland of the brain [8,9]. It can also be synthesized by ocular tissues, including the ciliary body and the retina [8,10]. Melatonin is involved in maintaining circadian rhythms and synchronizing various body functions, including maintenance of tissue metabolism, core body temperature, immune response, hormonal homeostasis, and blood pressure [10,11]. Two melatonin receptors, MT_1_ and MT_2_, and a putative MT_3_ receptor have been identified [12]. MT_1_ and MT_2_ are G protein-coupled receptors, while MT_3_, a detoxification cytosolic enzyme, has been suggested to belong to the reductase group, quinone reductase 2 (NQO2) [13]. The melatonin-binding site of MT_3_, purified from Syrian hamster kidney has been identified as the hamster homolog of the human NQO2 [14]. Melatonin receptors have been shown to express in the ciliary body of various species [9]. For example, MT_1_ receptor is present in rabbits, while MT_2_ receptor has been found in both rabbits and humans [15,16]. MT_3_ receptor has been demonstrated in African clawed frogs (Xenopus laevis) [17].

Physiologically, melatonin levels in the AH display diurnal variation [12,18], supporting its potential significance in regulating the circadian rhythm of AH dynamics and thereby IOP. While several studies have shown that administration of melatonin or its analogues lowers IOP in both humans [19,20,21] and animals [22,23,24,25], others have not observed this melatonin-induced IOP-lowering effect [26,27]. Recently, higher levels of melatonin in the AH were detected in patients with higher IOPs [28]. This controversy hinders the understanding of the precise functional significance of melatonin in regulating AH dynamics. This study aimed to determine the direct effects of melatonin on AH secretion across porcine ciliary epithelium, which is considered as a good animal model to mimic human conditions [29]. This was achieved by evaluating the cellular effects of melatonin on transepithelial Cl^−^ and fluid secretion, as well as gap junction permeability. The effects of various melatonin receptor antagonists on transepithelial Cl^−^ transport were also investigated.

## 2. Results

### 2.1. Melatonin Increases Short-Circuit Current (Isc) across Porcine Ciliary Epithelium

The effects of melatonin at different concentrations (1, 10, and 100 μM) on Isc across porcine ciliary epithelium were determined using a modified Ussing chamber. When administered bilaterally (i.e., applied to both stromal and aqueous sides), melatonin had no effect on Isc at 1 and 10 μM (Figure 1). However, at 100 μM, it significantly stimulated the Isc by 38 ± 6%.

Addition of melatonin only to the aqueous side triggered a similar concentration-dependent Isc stimulation (Figure 2A). The melatonin-induced Isc stimulation was only observed at 100 μM (40 ± 9%), but not at 1 or 10 μM. In contrast, no significant effect was observed at all concentrations tested when melatonin was added to the stromal side alone (Figure 2B).

The latency period after the administration of melatonin between the bilateral (42 ± 8 min) and aqueous (52 ± 9 min) applications were similar (*p* > 0.05, Student’s *t*-test). In addition, the melatonin-induced Isc stimulation observed in both aqueous and bilateral application was abolished by the addition of 1 mM niflumic acid (NFA), a non-selective Cl^−^ channel blocker. These results strongly suggested that melatonin stimulated Cl^−^ secretion across the porcine ciliary epithelium by enhancing Cl^−^ release from NPE cells into the posterior chamber.

### 2.2. Melatonin Stimulates Transepithelial Fluid Flow and Electrical Measurements

To confirm whether melatonin exerts any direct effect on fluid secretion, both Isc and transepithelial fluid movement were monitored simultaneously with a fluid flow chamber. The changes in Isc and fluid flow are summarized in Figure 3. Addition of melatonin (100 µM) to the aqueous side elicited a concomitant stimulation of Isc and fluid movement in the stromal-to-aqueous direction. A sustained increase in fluid flow was observed from the baseline value of 1.22 ± 0.14 µL/h to a maximum of 2.27 ± 0.29 µL/h after 90 min of melatonin treatment (Figure 3A).

### 2.3. Melatonin Enhances Gap Junction Permeability between PE and NPE Cells

As melatonin has been reported to modulate gap junction permeability [30,31], the effect of pre-treatment with heptanol, a non-selective gap junction blocker on Isc was investigated (Figure 4). Bilateral addition of heptanol (3.5 mM) abolished the baseline Isc and completely inhibited the subsequent melatonin-induced Isc stimulation across the porcine ciliary epithelium, suggesting that melatonin might increase Cl^−^ secretion by enhancing gap junction permeability.

The lucifer yellow (LY) dye transfer technique was employed to confirm whether melatonin has a direct effect on gap junction permeability. The diffusion of LY dye from PE to NPE cell couplet is shown in Figure 5. The relative fluorescence intensities (F ratios) of melatonin-treated and control groups were monitored for 30 min. The results showed that melatonin (100 μM) significantly enhanced the gap junction permeability across isolated PE-NPE cell couplets. At 15 min, the F ratios of the melatonin and the control groups were 2.64 ± 0.38 and 1.68 ± 0.15, respectively, indicating a 60% increase in gap junction permeability compared with the control. Similarly, pre-treatment with 100 μM melatonin for 45 min triggered a sustained increase in gap junction permeability. At 15 min, the F ratios of the melatonin-pretreated and the control groups were 3.06 ± 0.44 and 1.68 ± 0.15, respectively. The F ratios between the melatonin-pretreated and melatonin groups was not statistically significant at all time points (Figure 5).

### 2.4. Melatonin Stimulates Cl^−^ Secretion via MT_3_ Receptor

The gene expression level of melatonin receptors MT_1_, MT_2_, and MT_3_ was quantified in porcine NPE cells using RT-qPCR. The results demonstrated that MT_3_ was the most abundant isoform compared with MT_1_ and MT_2_ receptors (Figure 6).

In addition, effects of two melatonin antagonists, luzindole (an MT_1_/MT_2_ receptor antagonist) and prazosin (a putative MT_3_ receptor antagonist) were investigated. Both antagonists were pre-treated at the aqueous side of the ciliary epithelium prior to aqueous administration of melatonin (100 μM). The results indicated that luzindole (10 μM) did not affect the baseline Isc (Figure 7). Interestingly, luzindole enhanced the melatonin-induced Isc stimulation compared with application of melatonin alone (*p* < 0.05, Student’s *t*-test). The subsequent addition of 1 mM NFA significantly inhibited the melatonin-induced Isc stimulation, in either the presence or absence of luzindole pre-treatment.

The effects of prazosin (1.5 μM and 10 μM) on Isc are summarized in Figure 8. At 1.5 μM, prazosin had no significant effects on either baseline Isc or the subsequent melatonin-induced Isc stimulation. At 10 μM, prazosin pre-treatment did not affect the baseline Isc, but significantly inhibited the subsequent melatonin-induced Isc stimulation. Similar to luzindole, the subsequent addition of 1 mM NFA abolished Isc stimulation after prazosin pre-treatment (N = 5, *p* < 0.001, Student’s *t*-test).

## 3. Discussion

The results of this study suggest that melatonin, when administered to the aqueous side, elicits a concomitant stimulation of transepithelial Cl^−^ and fluid secretion across the isolated porcine ciliary epithelium, potentially increasing the rate of AH secretion. This finding was supported by the melatonin-induced facilitation of transcellular fluid movement from the PE to NPE in isolated cell couplets. The melatonin-induced stimulation of Cl^−^ secretion was mediated through MT_3_ receptors.

Similar to IOP, the level of melatonin in the AH displays a diurnal variation [12,18]. The concentration of melatonin in the AH has been shown to increase by 2–3 fold at night in rabbits [32] and humans [33,34], suggesting its potential contribution to the regulation of AH dynamics and IOP. Studies have shown that topical administration of melatonin or its analogues reduces IOP in rabbits and monkeys [22,25,35]. Human studies have also demonstrated that oral intake of melatonin or its analogues lowers IOP [19,20,21]. This is also supported by a study reporting that urinary melatonin levels in high-tension glaucoma patients were lower compared to those of control subjects [36]. Despite these reports, there has been no consensus on the pathway for melatonin to achieve IOP reduction. Based on a study using MT_1_ knock-out mice, it has been postulated that the MT_1_ receptor is primarily responsible for the ocular hypotensive effect [37]. Other studies have demonstrated that selective MT_2_ antagonists block the IOP-lowering effects of melatonin and its analogues [24]. In contrast, MT_3_ receptors have been proposed to trigger melatonin-induced IOP reduction [22,25]. To date, the precise cellular effects of melatonin in modulating AH secretion are still unconfirmed. The results of the current study showed that low concentrations of melatonin (1 or 10 μM) had no effect on Isc, when added to both stromal and aqueous sides, whereas a significant increase in Isc was observed at 100 μM. The increase in Isc may reflect a stimulation of net transepithelial Cl^−^ secretion across the ciliary epithelium [38]. Melatonin is highly lipophilic and can penetrate and/or diffuse across plasma membrane influencing both PE and NPE cells. However, administration of melatonin to the stromal side had a subtle effect on Isc (Figure 2B). On the contrary, melatonin, when applied to the aqueous side alone, produced similar responses to that of bilateral administration of melatonin with no additive effect (Figure 2A). This result strongly suggest that the effect of melatonin is primarily mediated in NPE cells [39]. This notion was further supported by the evidence that Isc stimulation by melatonin was completely blocked by either heptanol or NFA, indicating that solute uptake in PE cells was not rate limiting [38,40].

To confirm whether melatonin directly acts on active Cl^−^ and fluid secretion, the effect of melatonin on the transepithelial ion and fluid secretion across the intact porcine ciliary epithelium was determined. In agreement with the results obtained from electrical parameter measurements, we showed that melatonin, when applied to the aqueous side, triggered a ~80% increase in fluid secretion across the porcine ciliary epithelium. To the best of our knowledge, this is the first report of melatonin-stimulated fluid secretion concomitant with an increase in Isc. The melatonin-stimulated fluid secretion was further substantiated by the results demonstrating that melatonin (1) failed to stimulate Isc after heptanol pre-treatment and (2) enhanced transcellular fluid transfer from PE and NPE cells, as revealed by a sustained increase in LY dye transfer across isolated PE-NPE cell couplets. These results are in line with previous studies that melatonin plays a pivotal role in regulating gap junction permeability in various cell types [30,31,41,42,43]. Our finding suggests that melatonin may increase Cl^−^ secretion by enhancing gap junction permeability. This finding differs from a recent study in which melatonin and its analogue 5-MCA-NAT were shown to inhibit Cl^−^ efflux by rabbit NPE cells [39]. The exact reason for this discrepancy is not clear but could, at least in part, be explained by species differences [44]. It has been suggested that integrated transport mechanism of ions across the ciliary epithelium displays species variation [44]. In rabbits, AH secretion is primarily driven by HCO_3_^−^ transport, and to a lesser extent by Cl^−^ secretion [45,46,47]. The inhibition of Cl^−^ efflux in NPE cells may not necessarily lead to a suppression of AH production [48]. In contrast, Cl^−^ secretion has been shown to be the major driving force for AH secretion in animal species, including pigs, oxen, and humans [38,45,46,49]. It has been suggested that pig eye serves as a good animal model for studying human AH dynamics [5,29], partly because of their similarities to humans regarding anatomical structures, AH’s electrolyte composition, and responses to drugs [50,51,52,53]. Our current results were also in good agreement with a previous study, which demonstrated that melatonin stimulated Isc and Cl^−^ secretion in human colonic T84 cells [54].

It Is noted that the physiological concentration of melatonin in the AH was reported to be in the range of 4–48 pg/mL [32,33] to 0.5–46.7 ng/mL [28,55], depending on the experimental conditions and methodology adopted. The reported values were lower than the concentrations of melatonin used in the current study, although the chosen concentrations still fell within the working range used in the previous studies [39,41,54,56] with no cytotoxicity reported [57]. The discrepancy could also be attributed to the compartmentalization of melatonin in the ciliary epithelial cells [28] and the differences in metabolism, such as endogenous synthesis, secretion, and degradation of melatonin between physiological and experimental conditions.

In general, the functional significance of MT_3_ receptor is less well documented and characterized compared with MT_1_ and MT_2_ receptors [58]. Our results also provided the first evidence that melatonin potentially increases Cl^−^ secretion and AH secretion rate through MT_3_ receptors. We demonstrated that pre-treatment with 10 μM luzindole, a non-selective MT_1_/MT_2_ receptor antagonist, did not inhibit melatonin-induced Isc stimulation. This lack of inhibition exerted by luzindole suggested that the melatonin-induced response was not mediated through MT_1_/MT_2_ receptors. This finding was in parallel with a recent study in which melatonin-induced effect was enhanced, rather than prevented, by MT_1_/MT_2_ receptor antagonists [39]. In contrast, although pre-treatment with 1.5 μM prazosin, an MT_3_ receptor antagonist, had no effect on melatonin-induced Isc stimulation, at a higher concentration (10 μM), it abolished Isc stimulation by melatonin. It has been reported that prazosin (at 1.5 μM) was sufficient to inhibit the melatonin-induced response [39], but in the current study the inhibitory effect of prazosin was only observed at higher concentration. This difference may be explained by: (1) species difference in affinity to prazosin at the binding sites [59,60,61] and (2) difference in sample preparations as immortalized cultured NPE cells were used in the earlier study [39], whereas excised ciliary epithelium was used in the current study. The freshly isolated ciliary epithelium preparation may have vitreous humor attached to the ciliary epithelium, potentially hindering the access of melatonin to the target site(s), such as NPE cells [29]. Little is known about the presence of MT_3_ receptors in human ciliary epithelium. We found that the MT_3_ receptor was the most abundant isoform among all melatonin receptors in porcine ciliary epithelium, an animal model compared favorably with humans [29,62]. Additional work is warranted to determine the precise expression of melatonin receptors in human ciliary epithelium in the future.

Physiologically, melatonin has been demonstrated to trigger cellular effects via various signaling cascades, including cAMP, cGMP, PI3K/AKT, calmodulin, and phospholipase C [63,64,65,66,67,68]. In addition to intercellular gap junctions linking PE and NPE cells, it is likely that melatonin may affect Cl^−^ channels at the basolateral membrane of NPE cells. Although the identities of Cl^−^ channels are not fully understood [69], Cl^−^ efflux by NPE cells is a crucial step for AH secretion [6,7]. In our study, the melatonin-induced Isc stimulation was abolished by NFA administered to the aqueous side, suggesting that Cl^−^ channels might be a potential target for the observed response. Further studies are required to investigate the mechanistic actions of melatonin and its antagonists on the regulation of NPE-cell Cl^−^ channels.

This study’s in vitro findings suggest that melatonin may potentially stimulate Cl^−^ and, thereby, AH secretion, possibly through putative MT_3_ receptors (Figure 9). This result was in good agreement with a recent study that there was a good correlation between IOP and melatonin concentration in the AH [28]. Patients with ocular hypertension had a three-fold increase in melatonin concentration (46.63 ng/mL for IOP > 21 mmHg versus 14.62 ng/mL for IOP < 21 mmHg) [28]. Similarly, glaucomatous DBA/2J mice displayed a higher concentration of melatonin in the AH as compared to the control C57BL/6J mice [28]. Nevertheless, our finding differed from other reports of melatonin-induced IOP reduction in both experimental animals and humans [22,24]. As isolated porcine ciliary epithelium was used in this study, extraneous factors, including hydrostatic pressure, hormonal, and vascular influences, were excluded. In living animals, however, melatonin and its analogues may affect both AH secretion and drainage pathways concomitantly. This difference could reflect compensatory changes in outflow facility. For example, melatonin was found to stimulate voltage dependent Na^+^ current in human TM cells [70], potentially influencing the outflow facility and IOP. In addition, oxidative stress was reported to increase AH outflow resistance and IOP by affecting the integrity of TM cells [71,72,73]. As melatonin is a potent free radical scavenger, it is likely that melatonin may serve as an anti-oxidant by protecting TM cells against oxidative stress. Further studies are required to determine the precise functional significance of melatonin on the modulation of AH outflow resistance.

## 4. Materials and Methods

### 4.1. Transepithelial Electrical Measurements with Modified Ussing Chamber

Freshly enucleated porcine eyes were obtained from a local abattoir. For Ussing chamber experiments, a sector of the ciliary body epithelium was excised [38,74]. Similar to our previous study [29], the ciliary body was mounted onto the Ussing chamber with an exposed area of 0.10 cm^2^. Both hemi-chambers (i.e., stromal side facing the PE and aqueous side facing the NPE) of the Ussing chamber were filled with Ringer’s solution and then bubbled with a mixture of 95% O_2_ and 5% CO_2_ throughout the experiment. Drugs, including melatonin, niflumic acid (NFA), luzindole, and prazosin, were added either bilaterally or to the aqueous/stromal side only, as appropriate. As melatonin is sensitive to light [75], the experiment was conducted under dim light conditions. Transepithelial electrical parameters including the Isc were monitored with a dual voltage current clamp unit (World Precision Instruments, Sarasota, FL, USA).

### 4.2. Simultaneous Electrical and Fluid Flow Measurements with Custom-Built Chamber

Similar to Ussing chamber experiments, an intact annulus ring of porcine ciliary epithelium with iris attached was isolated for fluid flow measurements [29,76]. The whole ciliary body preparation was mounted between two hemi-chambers with an exposed area of 0.78 cm^2^. One hemi-chamber was connected to a bubbling reservoir to facilitate drug administration, while the other was connected to a 25 µL graduated capillary tube. Both sides of the chamber were filled with Ringer’s solution and bubbled with 95% O_2_ and 5% CO_2_ throughout the experiment. The spontaneous fluid flow was monitored every 15 min for 2–3 h by determining the changes in height of the water column in the capillary tube.

### 4.3. Measurement of Gap Junction Permeability with Lucifer Yellow (LY) Dye Transfer

As reported in our previous studies, LY dye diffusion was determined across freshly isolated porcine PE-NPE cell couplets [5,77]. For each PE-NPE cell couplet, a tight seal was formed between the micropipette and the plasma membrane of the PE cell (chosen as the donor cell). Melatonin was added to the bathing solution either 45 min before (i.e., melatonin pre-treatment) or at the time of membrane rupture. After membrane rupture, the rate of LY dye diffusion from the PE (donor) cell to the NPE (recipient) cell were determined. Images were captured every 30 s for 30 min by an inverted fluorescence microscope (Nikon Corp, Tokyo, Japan). F ratio (fluorescence intensity in NPE cell compared to that of the PE cell) was used for fluid flow analysis.

### 4.4. Reverse Transcription-Real Time Polymerase Chain Reaction (RT-qPCR)

As summarized previously [5], total RNA was extracted from the porcine ciliary epithelium by Qiagen RNeasy Micro Kit (Qiagen Co, Dusseldorf, Germany), quantified, and reverse transcribed to cDNA using a High-Capacity cDNA Reverse Transcription Kit (Applied Biosystems, Waltham, MA, USA). Real time-qPCR was conducted using the LightCycler 480 SYBR Green I Master (Roche Applied Science, Penzberg, Germany) with primers specific for the target gene MT_1_ (forward primer: 5′-CTCGCGCTCATCCTCATCTT-3′; reverse primer: 5′-TTCCTGCGTTCCTCAGCTTC-3′), MT_2_ (forward primer: 5′-GAGCATGTTCGTGGTGTTCG-3′; reverse primer: 5′-CCTGCGGAAGTTCTGGTTCA-3′), MT_3_ (forward primer: 5′-TCAGGAGGCTGATCTGGTGA-3′; reverse primer: 5′-GACGGCCAGTTTACCCTTGA-3′), and the internal reference gene GAPDH. qPCR was performed in 96-well plates on the ROCHE LightCycler 480 (Roche Applied Science). A total reaction volume of 10 μL qPCR mixture containing 5 μL of 2 × Taq PCR Master Mix, 1 μL of sterile water, 2 μL of cDNA template, and 1 μL of 10 μM primers (forward and reverse primers, respectively) was used. The thermal cycling conditions were: 95 °C for 5 min, followed by 40 cycles of 95 °C for 30 s, 61 °C for 30 s and 72 °C for 30 s. A melting-curve analysis was performed to rule out primer–dimer formation and non-specific product amplification. Data were analyzed using Light LC480 software.

### 4.5. Solutions and Pharmacological Agents

All chemicals were purchased from Sigma-Aldrich. HEPES-buffered Ringer’s solution, composed of (in mM) 113.0 NaCl, 21.0 NaHCO_3_, 10.0 HEPES, 7.5 D-Glucose, 4.56 KCl, 1.4 CaCl_2_, 1.0 L-Glutathione (reduced), 1.0 Na_2_HPO_4_, and 0.6 MgSO_4_ was used as the bathing solution for all experiments. The pH and osmolarity of the solution were adjusted to 7.4 and 300 mOsm/kg, respectively. For LY dye diffusion, the pipette solution contained 120 mM N-Methyl-D-glucamine (NMDG), 110 mM L-aspartic acid, 25 mM NaCl, 0.38 mM CaCl_2_, and 12 mM HEPES with 1 mg/mL LY CH, lithium salt (Invitrogen, Waltham, MA, USA). All drugs used in the study, including melatonin, luzindole, prazosin, and NFA (except heptanol), were dissolved in dimethyl sulfoxide (DMSO). The final concentration of DMSO was adjusted to <0.1% in the bathing solution.

### 4.6. Statistical Analysis

All data were presented as mean ± SEM. One-way repeated measures ANOVA or Student’s *t*-test were used to analyze the data. *p* < 0.05 was considered statistically significant.

## 5. Conclusions

Currently, lowering IOP is the only available clinical intervention known to delay the onset and progression of glaucomatous blindness. The results of this study indicate that melatonin increases gap junction permeability between PE and NPE cells, facilitating Cl^−^ and fluid secretion across the porcine ciliary epithelium via MT_3_ receptors. Whether or not melatonin contributes to the regulation of both AH secretion and drainage, and their relationship to IOP, awaits further investigation.

## Figures and Tables

**Figure 1 ijms-24-05789-f001:**
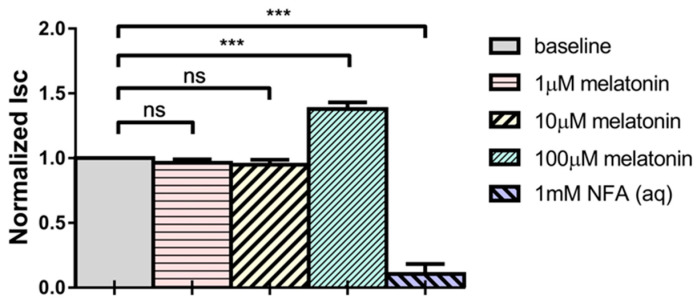
Effects of melatonin, when added to stromal and aqueous sides simultaneously, on short-circuit current (Isc) (N = 6, ns: not statistically significant, *** *p* < 0.001, one-way repeated measures ANOVA).

**Figure 2 ijms-24-05789-f002:**
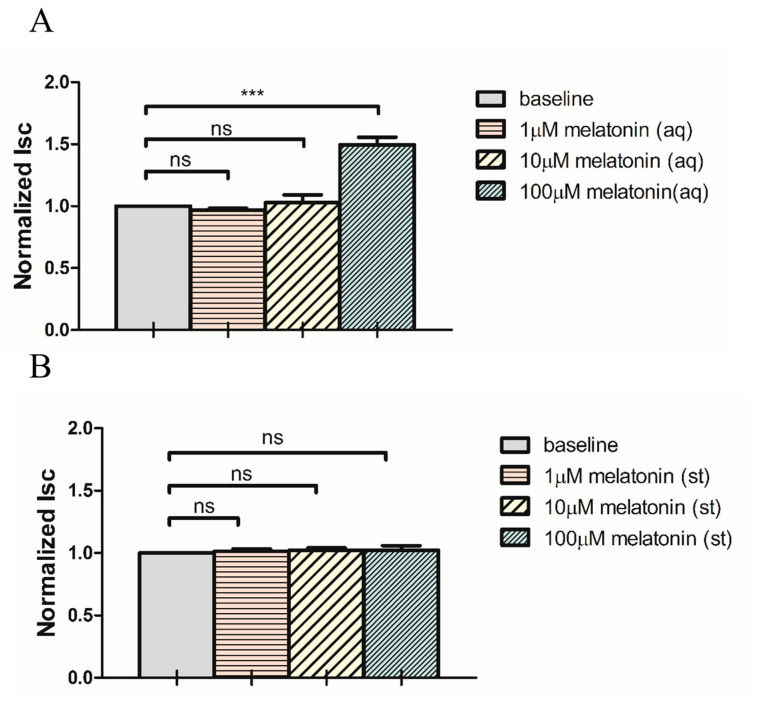
Effects of melatonin, when added to (**A**) aqueous (N = 11) and (**B**) stromal (N = 4) sides only, on short-circuit current (Isc) (ns: not statistically significant, *** *p* < 0.001, one-way repeated measures ANOVA).

**Figure 3 ijms-24-05789-f003:**
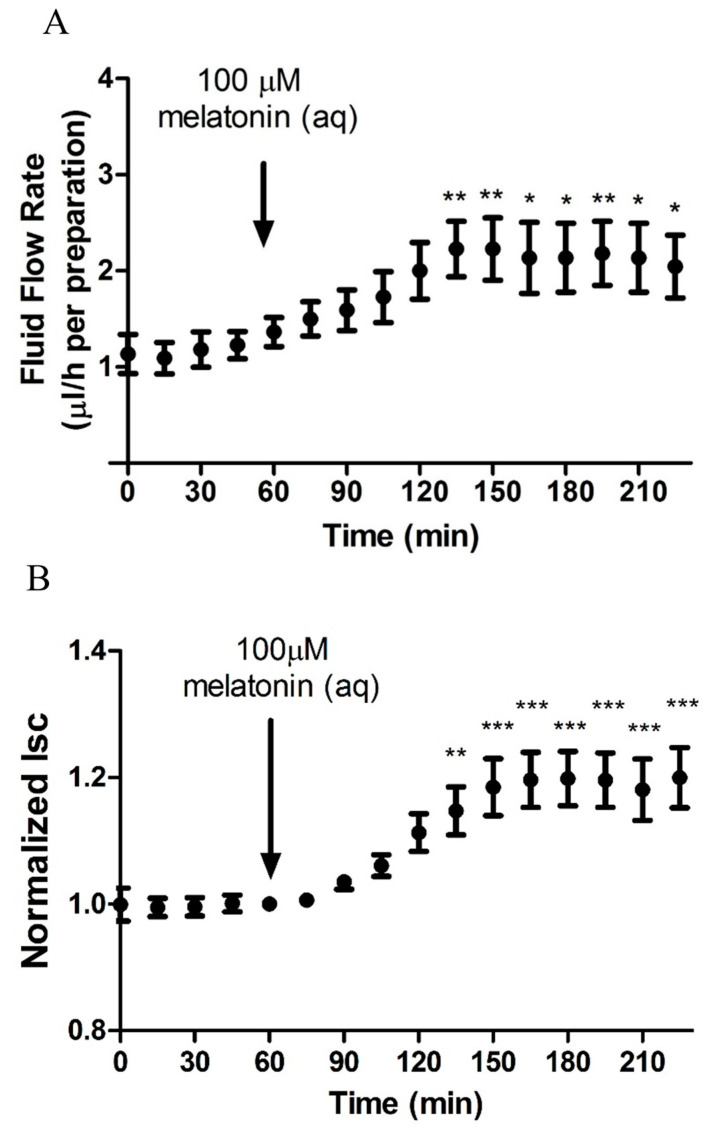
Effects of 100 µM melatonin on (**A**) transepithelial fluid flow and (**B**) Isc across porcine ciliary epithelium using fluid flow chamber. Data points were plotted with mean ± SEM. Melatonin was administered to the aqueous (aq) side of the ciliary body. Average fluid flow rate of the 1-h period prior to melatonin administration was regarded as the baseline. Isc just before the administration of melatonin was regarded as the baseline Isc (N = 11, * *p* < 0.05, ** *p* < 0.01, *** *p* < 0.001, one-way repeated measures ANOVA).

**Figure 4 ijms-24-05789-f004:**
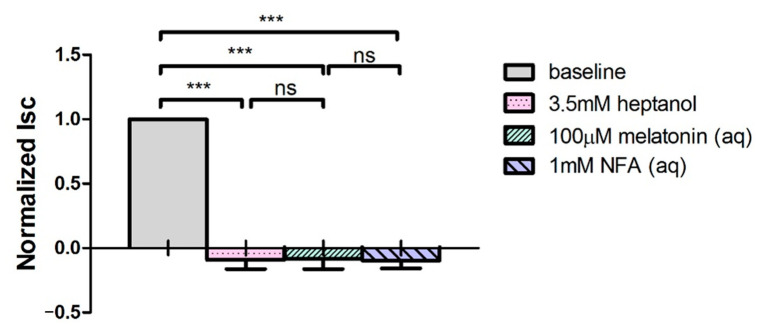
Heptanol pre-treatment inhibited melatonin-induced Isc stimulation (N = 8, ns: not statistically significant, *** *p* < 0.001, one-way repeated measures ANOVA).

**Figure 5 ijms-24-05789-f005:**
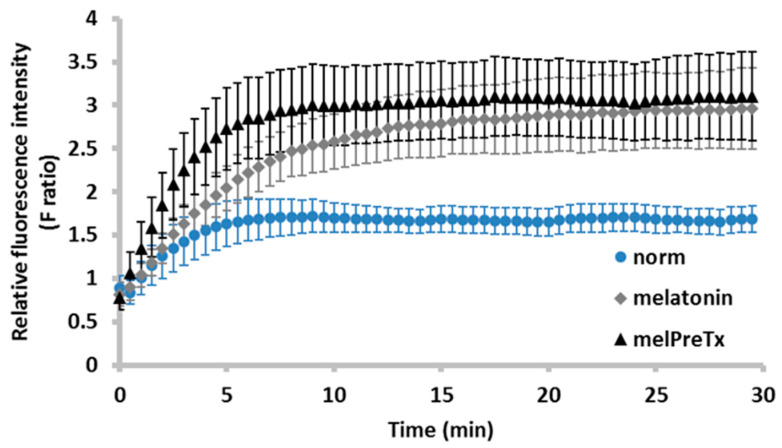
Relative fluorescence intensity (F ratio) over time with melatonin treatment. norm: normal as control (N = 10); melatonin: 100 μM melatonin (N = 9); melPreTx: 100 μM melatonin pretreatment for 45 min (N = 7).

**Figure 6 ijms-24-05789-f006:**
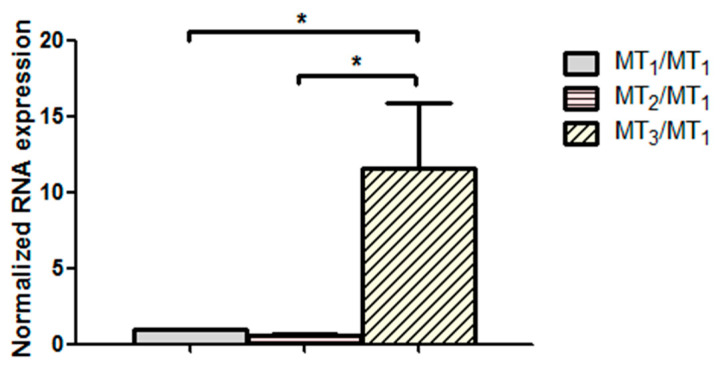
Quantitative characterization of melatonin receptors MT_1_, MT_2_, and MT_3_ in porcine ciliary epithelium (normalized with MT_1_) (N = 4, * *p* < 0.05, one-way ANOVA).

**Figure 7 ijms-24-05789-f007:**
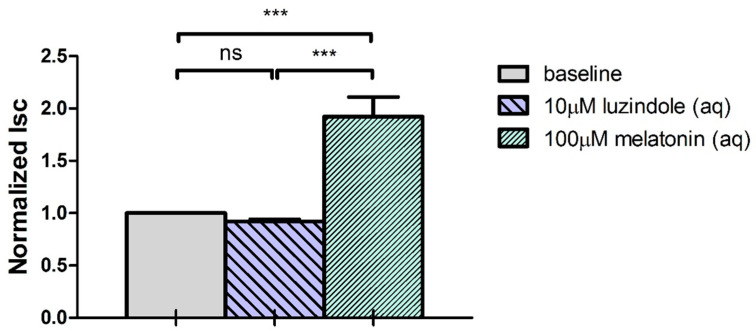
MT_1_/MT_2_ antagonist luzindole pre-treatment on melatonin-induced Isc response (N = 10, ns: not statistically significant, *** *p* < 0.001, one-way repeated measures ANOVA).

**Figure 8 ijms-24-05789-f008:**
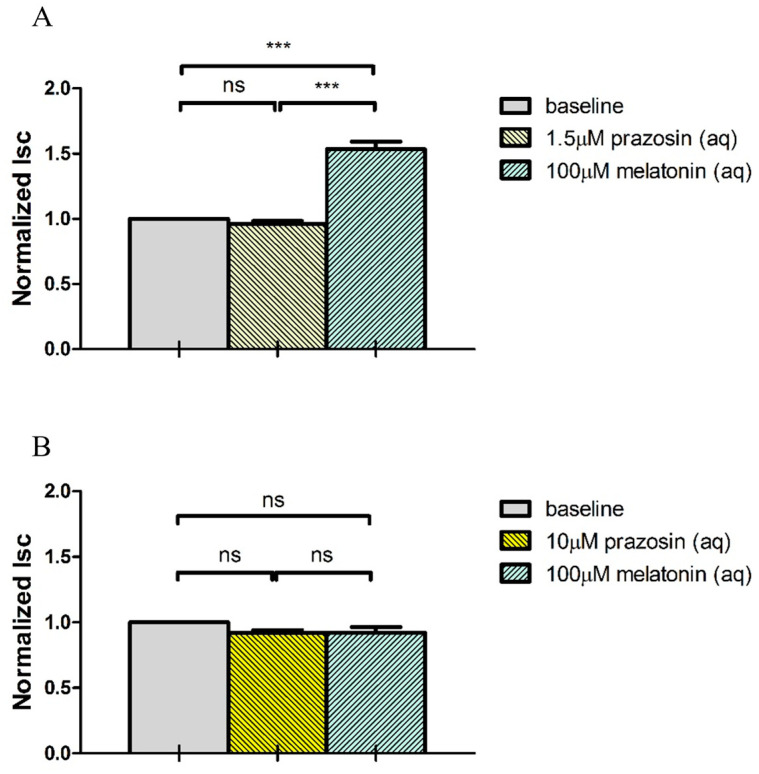
MT_3_ antagonist prazosin pre-treatment in the concentration of (**A**) 1.5 μM (N = 10) and (**B**) 10 μM (N = 10) on melatonin-induced Isc response (ns: not statistically significant, *** *p* < 0.001, one-way repeated measures ANOVA).

**Figure 9 ijms-24-05789-f009:**
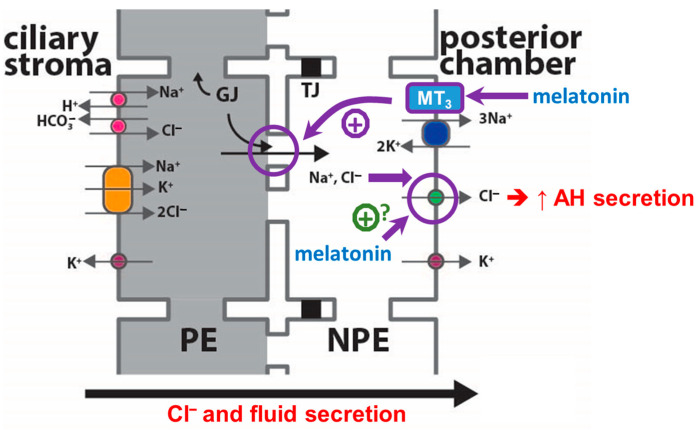
Schematic diagram showing the proposed cellular effects of melatonin on aqueous humor (AH) secretion. NPE: non-pigmented ciliary epithelium; PE: pigmented ciliary epithelium; Isc: short-circuit current; TJ: tight junction; GJ: gap junction; MT_3_: MT_3_ melatonin receptor.

## Data Availability

The data presented in this study are available on request from the corresponding author.

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
