# Peer review of "Regulation of Aqueous Humor Secretion by Melatonin in Porcine Ciliary Epithelium"

_ijms, 2023, doi:10.3390/ijms24065789_

Round 1
Reviewer 1 Report
The study conducted by Li and colleagues is well-designed and structured and provides important insights into the mechanisms of how melatonin regulates aqueous secretion in the porcine ciliary epithelium. They first administered a high dosage of melatonin (100um) to both sides of the epithelium and observed a significant increase in the short-circuit current (Isc). They later found that this increase was only effective when melatonin was applied to the aqueous side of the epithelium.
In addition to measuring Isc, the researchers also simultaneously monitored fluid flow and found that the increases in Isc and fluid flow were concomitant. Lastly, they provided evidence that melatonin may increase Cl-secretion by enhancing gap junction permeability, and this effect is mediated through MT3 receptors.
It is important to note that while the study was conducted in the porcine ciliary epithelium, the findings have potential implications for understanding how melatonin regulating aqueous humor secretion in other species, including humans. However, there are a few points that could be addressed to further improve the soundness of the results and the clarity of the article.
Firstly, the study reports that melatonin increases Cl-secretion by enhancing gap junction permeability and mediated through MT3 receptors. More background information on the MT3 receptor subtype would be helpful to justify the conclusions drawn in this study. For example, are there any reports showing the presence of MT3 receptors in human ciliary PE or NPE cells? If MT3 is rare in human ciliary PE and NPE cells, how would the authors justify their conclusions? It is important to note that this study almost confirmed that MT1 and MT2 do not participate in Cl-secretion and thus aqueous secretion.
Secondly, the study uses a high dosage of melatonin (100um) to determine its effects on aqueous humor secretion in the porcine ciliary epithelium. It would be helpful to know what the normal melatonin levels are in both human and porcine aqueous humor, as this would provide a context for interpreting the results. To the best of our knowledge, the normal melatonin levels in aqueous humor are below 20pg/ml. This difference in concentration raises questions about the potential toxicity of high dosages of melatonin and the physiological relevance of the findings.
Author Response
We thank the Reviewer for the encouragement and constructive comments that help improve the quality of the manuscript. In response to the valuable suggestions, we have provided the following point-by-point response to each of the comments. The corresponding changes are also made in the revised manuscript for easy reference.
Thank you for the excellent comment. This is fundamentally important. To my humble understanding, there is no evidence demonstrating whether human PE/NPE cells express MT3 receptors. In general, the characterization of MT3 receptor is not well documented compared with MT1 and MT2 receptors [1]. In the revised manuscript, we have included more description of MT3 receptor in the Introduction and Discussion. We have also emphasized that …“little is known about the presence of the MT3 receptor in human ciliary epithelium. We found that MT3 receptor was the most abundant isoform among all melatonin receptors in porcine ciliary epithelium, an animal model compared favourably with humans [2, 3]. Additional work is warranted to determine the precise expression of melatonin receptors in human ciliary epithelium in the future”.
In the literature, the physiological concentration of melatonin in the aqueous humor was reported to be ranging from 4 - 48 pg/mL [4, 5] to 0.5 - 46.7 ng/mL [6, 7], depending on the experimental conditions and methodology adopted. These values were lower than the melatonin concentrations used in the current study, although the chosen concentrations still fell within the working range used in the previous studies [8-11] with no reported cytotoxicity [12]. The discrepancy could also reflect the compartmentalization of melatonin in the ciliary epithelial cells [7], and the differences in metabolism such as endogenous synthesis, secretion, and degradation of melatonin between physiological and experimental conditions.
References:
- Ekmekcioglu, C., Melatonin receptors in humans: biological role and clinical relevance. Biomed Pharmacother 2006, 60, (3), 97-108.
- Cheng, A. K.; Civan, M. M.; To, C. H.; Do, C. W., cAMP stimulates transepithelial short-circuit current and fluid transport across porcine ciliary epithelium. Invest Ophthalmol Vis Sci 2016, 57, (15), 6784-6794.
- Pelis, R. M.; Shahidullah, M.; Ghosh, S.; Coca-Prados, M.; Wright, S. H.; Delamere, N. A., Localization of multidrug resistance-associated protein 2 in the nonpigmented ciliary epithelium of the eye. J Pharmacol Exp Ther 2009, 329, (2), 479-85.
- Chiquet, C.; Claustrat, B.; Thuret, G.; Brun, J.; Cooper, H. M.; Denis, P., Melatonin concentrations in aqueous humor of glaucoma patients. American journal of ophthalmology 2006, 142, (2), 325-327.
- Liu, J. H.; Dacus, A. C., Endogenous hormonal changes and circadian elevation of intraocular pressure. Invest Ophthalmol Vis Sci 1991, 32, (3), 496-500.
- Martin, X. D.; Malina, H. Z.; Brennan, M. C.; Hendrickson, P. H.; Lichter, P. R., The ciliary body--the third organ found to synthesize indoleamines in humans. Eur J Ophthalmol 1992, 2, (2), 67-72.
- Alkozi, H.; Sanchez-Naves, J.; de Lara, M. J.; Carracedo, G.; Fonseca, B.; Martinez-Aguila, A.; Pintor, J., Elevated intraocular pressure increases melatonin levels in the aqueous humour. Acta Ophthalmol 2017, 95, (3), e185-e189.
- Kojima, T.; Mochizuki, C.; Mitaka, T.; Mochizuki, Y., Effects of melatonin on proliferation, oxidative stress and Cx32 gap junction protein expression in primary cultures of adult rat hepatocytes. Cell Struct Funct 1997, 22, (3), 347-56.
- Chan, H. C.; Lui, K. M.; Wong, W. S.; Poon, A. M., Effect of melatonin on chloride secretion by human colonic T84 cells. Life Sci 1998, 62, (23), 2151-8.
- Dortch-Carnes, J.; Tosini, G., Melatonin receptor agonist-induced reduction of SNP-released nitric oxide and cGMP production in isolated human non-pigmented ciliary epithelial cells. Exp Eye Res 2013, 107, 1-10.
- Huete-Toral, F.; Crooke, A.; Martinez-Aguila, A.; Pintor, J., Melatonin receptors trigger cAMP production and inhibit chloride movements in nonpigmented ciliary epithelial cells. J Pharmacol Exp Ther 2015, 352, (1), 119-28.
- Zhelev, Z.; Ivanova, D.; Bakalova, R.; Aoki, I.; Higashi, T., Synergistic Cytotoxicity of Melatonin and New-generation Anticancer Drugs Against Leukemia Lymphocytes But Not Normal Lymphocytes. Anticancer Res 2017, 37, (1), 149-159.
Reviewer 2 Report
Authors should depict a schematic cartoon that informs the readership all details about the experiments. How would they introduce a soluble protein in one side and eliminate diffusion or how would the diffusion not matter in their experiments is unclear. Authors have used 100 micromolar melatonin, is it physiological? What is the physiological concentration of melatonin in pig eyes? Do the light exposure had an effect (PMID: 30074278)? Some of these questions should factor in their study design or be explained.
Author Response
We sincerely thank the Reviewer for the time and great effort in reviewing our manuscript. Following the Reviewer’s suggestion, we have added a schematic diagram (Figure 9) to illustrate the proposed effects of melatonin in driving aqueous humor secretion.
In the revised manuscript, we have also mentioned that melatonin is highly lipophilic and can penetrate and/or diffuse across plasma membrane influencing both PE and NPE cells. We observed that stromal application of melatonin had no effect on Isc (Figure 2B), while aqueous administration of melatonin elicited similar responses, e.g. magnitude of Isc stimulation and latency period, as compared to bilateral administration of melatonin (with no additive effects). This result strongly suggests that the effect of melatonin is primarily mediated in NPE cells. This notion was further supported by the evidence that Isc stimulation by melatonin was completely inhibited by either heptanol or NFA, indicating that solute uptake in PE cells was not rate limiting [1, 2].
As mentioned above, the physiological concentration of melatonin in the aqueous humor is expected to be in the range from 4 - 48 pg/mL [3, 4] to 0.5 - 46.7 ng/mL [5, 6], depending on the experimental conditions and methods adopted. These values were lower than the melatonin concentrations used in the current study, although the chosen concentrations still fell within the working range used in the previous studies [7-10] with no reported cytotoxicity [11]. The discrepancy could be attributed to the compartmentalization of melatonin in the ciliary epithelial cells [6], and the differences in metabolism such as endogenous synthesis, secretion, and degradation of melatonin between physiological and experimental conditions.
We have also highlighted in the revised manuscript that the experiments were conducted under dim light condition, as melatonin is light sensitive. Thank you.
References
- Do, C. W.; To, C. H., Chloride secretion by bovine ciliary epithelium: a model of aqueous humor formation. Invest Ophthalmol Vis Sci 2000, 41, (7), 1853-60.
- Do, C. W.; Kong, C. W.; To, C. H., cAMP inhibits transepithelial chloride secretion across bovine ciliary body/epithelium. Invest Ophthalmol Vis Sci 2004, 45, (10), 3638-43.
- Chiquet, C.; Claustrat, B.; Thuret, G.; Brun, J.; Cooper, H. M.; Denis, P., Melatonin concentrations in aqueous humor of glaucoma patients. American journal of ophthalmology 2006, 142, (2), 325-327.
- Liu, J. H.; Dacus, A. C., Endogenous hormonal changes and circadian elevation of intraocular pressure. Invest Ophthalmol Vis Sci 1991, 32, (3), 496-500.
- Martin, X. D.; Malina, H. Z.; Brennan, M. C.; Hendrickson, P. H.; Lichter, P. R., The ciliary body--the third organ found to synthesize indoleamines in humans. Eur J Ophthalmol 1992, 2, (2), 67-72.
- Alkozi, H.; Sanchez-Naves, J.; de Lara, M. J.; Carracedo, G.; Fonseca, B.; Martinez-Aguila, A.; Pintor, J., Elevated intraocular pressure increases melatonin levels in the aqueous humour. Acta Ophthalmol 2017, 95, (3), e185-e189.
- Kojima, T.; Mochizuki, C.; Mitaka, T.; Mochizuki, Y., Effects of melatonin on proliferation, oxidative stress and Cx32 gap junction protein expression in primary cultures of adult rat hepatocytes. Cell Struct Funct 1997, 22, (3), 347-56.
- Chan, H. C.; Lui, K. M.; Wong, W. S.; Poon, A. M., Effect of melatonin on chloride secretion by human colonic T84 cells. Life Sci 1998, 62, (23), 2151-8.
- Dortch-Carnes, J.; Tosini, G., Melatonin receptor agonist-induced reduction of SNP-released nitric oxide and cGMP production in isolated human non-pigmented ciliary epithelial cells. Exp Eye Res 2013, 107, 1-10.
- Huete-Toral, F.; Crooke, A.; Martinez-Aguila, A.; Pintor, J., Melatonin receptors trigger cAMP production and inhibit chloride movements in nonpigmented ciliary epithelial cells. J Pharmacol Exp Ther 2015, 352, (1), 119-28.
- Zhelev, Z.; Ivanova, D.; Bakalova, R.; Aoki, I.; Higashi, T., Synergistic Cytotoxicity of Melatonin and New-generation Anticancer Drugs Against Leukemia Lymphocytes But Not Normal Lymphocytes. Anticancer Res 2017, 37, (1), 149-159.
Round 2
Reviewer 1 Report
No further questions.